# P2Y_12_ Inhibition beyond Thrombosis: Effects on Inflammation

**DOI:** 10.3390/ijms21041391

**Published:** 2020-02-19

**Authors:** Alexandre Mansour, Christilla Bachelot-Loza, Nicolas Nesseler, Pascale Gaussem, Isabelle Gouin-Thibault

**Affiliations:** 1Université de Paris, Innovative Therapies in Haemostasis, INSERM 1140, F-75006 Paris, France; alexandre.mansour@chu-rennes.fr (A.M.); pascale.gaussem@parisdescartes.fr (P.G.); 2CHU de Rennes, Department of Anesthesiology Critical Care Medicine and Perioperative Medicine, F-35000 Rennes, France; nicolas.nesseler@chu-rennes.fr; 3INSERM, CIC 1414 (Centre d’Investigation Clinique de Rennes), Université de Rennes, CHU de Rennes, F-35000 Rennes, France; 4INRA, INSERM, Institut NUMECAN–UMR_A 1341, Université de Rennes, CHU de Rennes, UMR_S 1241, F-35000 Rennes, France; 5AP-HP, Hôpital Européen Georges Pompidou, Department of Biological Hematology, F-75015 Paris, France; 6CHU de Rennes, Department of Biological Hematology, F-35000 Rennes, France

**Keywords:** platelets, P2Y_12_, inflammation, hemostasis, sepsis, cancer, leukocytes, antiplatelet agents, asthma, atherosclerosis

## Abstract

The P2Y_12_ receptor is a key player in platelet activation and a major target for antithrombotic drugs. The beneficial effects of P2Y_12_ receptor antagonists might, however, not be restricted to the primary and secondary prevention of arterial thrombosis. Indeed, it has been established that platelet activation also has an essential role in inflammation. Additionally, nonplatelet P2Y_12_ receptors present in immune cells and vascular smooth muscle cells might be effective players in the inflammatory response. This review will investigate the biological and clinical impact of P2Y_12_ receptor inhibition beyond its platelet-driven antithrombotic effects, focusing on its anti-inflammatory role. We will discuss the potential molecular and cellular mechanisms of P2Y_12_-mediated inflammation, including cytokine release, platelet–leukocyte interactions and neutrophil extracellular trap formation. Then we will summarize the current evidence on the beneficial effects of P2Y_12_ antagonists during various clinical inflammatory diseases, especially during sepsis, acute lung injury, asthma, atherosclerosis, and cancer.

## 1. Introduction

Nucleotides are universal extracellular signaling molecules, acting as intercellular or autocrine messengers that can be passively released by injured cells or secreted by specific mechanisms. Among their countless physiological or pathological functions, purinergic signaling can regulate hemostasis, thrombosis, and inflammation through the costimulation of various cell types, including platelets, leukocytes, endothelial, and vascular smooth muscle cells [1].

The specific plasma membrane receptors for nucleotides are called P2 receptors and are divided into two subgroups: P2X ligand-gated ion channels and P2Y G-protein (guanine nucleotide-binding protein)-coupled receptors. Eight P2Y receptors have been identified and divided according to their preferred agonist: adenine nucleotides (ADP and ATP) for P2Y1, P2Y11, P2Y_12_, and P2Y13 receptors; uracil nucleotides (UDP and UTP) for P2Y4 and P2Y6; adenine and uracil nucleotides for P2Y2; and UDP and UDP-glucose for P2Y14 [2,3,4,5]. 

Adenine nucleotide mediated platelet activation is a critical mechanism in both physiological and pathological hemostasis (including thrombosis), and it involves three platelet P2 receptors (P2Y1, P2Y_12_, and P2X1, the latter one being an ATP channel). Among the P2Y ADP receptors, P2Y_12_ is a key receptor and the unique P2 target for clinically approved antiplatelet drugs (herein called P2Y_12_ inhibitors) [6].

Besides their hemostatic capacities, platelets play an emerging and significant role in regulating inflammatory and immune response. Indeed, they are able to interact with immune cells through membrane exposure of P-selectin and CD40 (cluster of differentiation 40) ligand, and to release inflammatory mediators (cytokines, chemokines) [7,8]. Platelets are also involved in tumorigenesis and in the modulation of tumor microenvironment [9]. Moreover, P2Y_12_ is expressed in other cell types than platelets, including cancer, immune and vascular cells, and binding of ADP to P2Y_12_ might activate leukocytes and dendritic cells [2]. 

Therefore, together with antithrombotic effects, P2Y_12_ inhibitors might also have a role in modulation of inflammation.

After a brief description of P2Y_12_ function and its current clinical inhibitors, we will review the effects of P2Y_12_ inhibition on molecular and cellular mechanisms of inflammation and the current evidence for P2Y_12_ antagonism in clinical inflammatory diseases and syndromes.

## 2. P2Y_12_ Receptors

### 2.1. Structure of P2Y_12_ Receptors

The P2Y_12_ receptor was originally described as a platelet ADP receptor inhibited by thienopyridine antiplatelet agents ticlopidine and clopidogrel. It was further identified by cloning in 2001 by Hollopeter and Zhang and maps to chromosome 3q25.1 [10,11]. 

P2Y_12_ is a Gi-coupled seven-transmembrane domain receptor composed of 342 amino acids with a molecular weight of 39kDa. Its structure consists of a seven-transmembrane (7-TM) bundle of α-helices connected by three intracellular and three extracellular loops (EL), and a carboxy-terminal helix (H8) that is parallel to the membrane bilayer on the cytoplasmic side [12]. It contains four extracellular cysteines at positions 17, 97, 175, and 270 that form two disulfide bridges (between the *N*-terminal domain and EL3 and between EL1 and EL2) which are important sites for receptor expression and potential target for active metabolites of thienopyridines [2,13,14].

### 2.2. Expression of P2Y_12_ Receptor

The P2Y_12_ receptor was originally sought to be exclusively expressed on platelets, with about 400 copies of the P2Y_12_ receptor per cell, and to a lesser extent in subregions of the brain [11,15]. However, further studies demonstrated that P2Y_12_ has a wider tissue distribution, being expressed and functional in microglial cells [16], vascular smooth muscle cells [17,18], and on several immune cells including dendritic cells [19], mast cells [20], eosinophils [21], monocytes [22], lymphocytes [23,24] and macrophages [25]. The P2Y_12_ receptor expression has been also recently reported in osteoclasts [26] as well as in brain and breast cancer cell lines [27]. P2Y_12_ is not expressed by human endothelial cells of aortic, cerebral or cardiac origin [28]. Still, the expression and function of P2Y_12_ in other cell types remain poorly investigated. Although P2Y_12_ is expressed on the plasma membrane of resting platelets, an inducible pool of P2Y_12_ can also be exposed upon platelet activation by strong agonists [29].

### 2.3. Role of P2Y_12_ Rreceptor in Platelet Activation Pathways

Among extracellular nucleotides, ADP (adenosine 5′-diphosphate) plays a key role in platelet function and thrombus formation (Figure 1). It is a weak platelet agonist as it only induces reversible responses, including shape change and reversible aggregation. Platelets express two G protein-coupled receptors for ADP: P2Y1, which is coupled to Gq (G-protein q), and P2Y_12_, which is coupled to Gi (G-protein i) [3]. Despite the fact that ADP alone is not able to induce the release of platelet dense granules, binding of ADP to its platelet receptors amplifies and sustains the secretion and aggregation induced by other strong agonists [3]. Hence, P2Y_12_ plays an important role in the stabilization of platelet aggregates induced by other agonists such as thrombin and thromboxane A2 (TXA2), as well as in thrombogenesis in vivo [3,13,18,30,31]. This central position explains both the clinical benefit of P2Y_12_ antagonists and the bleeding diathesis observed in P2Y_12_ deficient patients [32,33].

Concomitant stimulation of P2Y_1_ and P2Y_12_ receptors by ADP is required to generate normal ADP-induced platelet aggregation [2,4]. Binding of ADP to P2Y_1_ initiates shape change and early reversible aggregation by mobilization of calcium ions from internal stores through the stimulation of phospholipase C and formation of inositol 1,4,5-trisphosphate (IP3) [5,13,34]. P2Y_12_ activates a G αi2 G protein subtype that mediates the inhibition of adenylate cyclase and therefore cyclic adenosine monophosphate (cAMP) production, leading to impaired protein kinase A (PKA) activation and a subsequent inhibition of downstream effectors such as vasodilator-stimulated phosphoprotein (VASP) [5,13,35]. This cAMP inhibition is not sufficient to induce platelet aggregation by itself but can promote it [36]. VASP phosphorylation state can be used to monitor the effects of anti P2Y_12_ therapy [37,38]. Binding of ADP to P2Y_12_ also induces an activation of two isoforms of phosphoinositide-3-kinase-(PI3K) - p110β and p110γ - through the recruitment of βγ subunits of Gi, which is critical for integrin αIIbβ3 (GpIIbIIIa) activation and stabilization of platelet aggregates, especially induced by thrombin or TXA2 [13,39,40,41,42]. This PI3K activation also amplifies platelet secretion induced by other agonists [13,43]. Finally, P2Y_12_ supports thrombin generation by amplifying membrane exposition of phosphatidylserine, platelet-derived microparticle formation and collagen-induced exposure of tissue factor (TF). Moreover, it contributes to leukocyte activation induced by surface P-selectin exposure and formation of platelet-leukocyte aggregates [13,44,45,46,47,48]. Formation of coated-platelets—which are a subpopulation of platelets characterized by surface retention of procoagulant proteins, expression of phosphatidylserine and strong prothrombinase activity—is dependent on ADP-induced P2Y_12_ activation [49,50,51].

### 2.4. Role of Non-Platelet P2Y_12_ Receptors

Vascular smooth muscle cells (VSMCs) play a key role in the physiological functions of blood vessels, such as vasoconstriction, vasodilatation and extracellular matrix production. They are also involved in the pathogenesis of vascular diseases, especially atherosclerosis, vascular inflammation and restenosis following angioplasty [52]. P2Y_12_ receptor expression and function in ADP-induced vessel contraction was first described in human internal mammary artery SMCs in 2004 [18]. Activation of P2Y_12_ is thought to induce an inflammatory state in VSMCs and correlates with atherosclerotic plaque instability [17,52,53,54]. It is associated with increased monocyte chemoattractant protein 1 (MCP-1) expression and monocyte adhesion [54]. P2Y_12_ is also implicated in the migration VSMCs through cAMP/PKA signaling pathway associated with actin disassembly and therefore an increase in VSMC motility and migration [55]. 

P2Y_12_ is also functionally expressed in microglial cells and can play a role in their activation [16] and in microglia-neuron communications and microglial neuroprotection [56]. Thus, as microglial cells are the primary innate immune cells of the brain and play an important role in the pathophysiology of many brain-based conditions, an implication of P2Y_12_ may be expected in various diseases including multiple sclerosis, Alzheimer’s disease, traumatic brain and spinal cord injury, and brain cancers [55].

Dendritic cells (DC) are considered the most efficient antigen presenting cells and are able to regulate adaptive immunity by inducing naïve T cell activation and effector differentiation [57]. P2Y_12_ receptor is expressed in murine DCs and its activation enhances specific T cell activation by increasing antigen endocytosis [19]. P2Y_12_ is also expressed in human DCs and it has been demonstrated that inhibition of P2Y_12_ mediated PI3K activation induces an immunosuppressive effect on DCs by decreasing antigen uptake [58].

To date, only few studies have evaluated the role of P2Y_12_ receptor in the purinergic signaling in leukocytes. High amounts of the P2Y_12_ receptor mRNA were previously described in lymphocytes as well as in CD34+ progenitor cells [24]. In a clinical study involving cardiologic patients, Diehl et al. showed that P2Y_12_ mRNA was expressed in leukocytes obtained from leukapheresis and that P2Y_12_ inhibition by clopidogrel decreased leukocyte CD11b (Mac-1) expression [59]. The P2Y_12_ expression was also reported in human eosinophils in which P2Y_12_ activation induced the release of eosinophil peroxidase [21] that could be prevented by clopidogrel [60]. Activation of P2Y_12_ in macrophages induced cell spreading with formation of lamellipodia, and P2Y_12_ inhibition alleviated chemotaxis [25]. Micklewright et al. demonstrated in human THP-1 monocytic cells that P2Y_12_ was expressed and positively regulated P2Y6-mediated intracellular Ca^2+^ signaling through suppression of adenylate cyclase activity [22]. Finally, Vemulapalli et al. recently showed that functional P2Y_12_ was expressed by T lymphocytes and that it exerted effects on biological responses of T cells when stimulated [23].

Taken together, these results suggest that P2Y_12_ inhibitors might target immune cell function and contribute to the modulation of inflammatory and immune responses.

Even though P2Y_12_ inhibitors are being explored as potential anti-cancer agents, expression of P2Y_12_ receptor in cancer cells has only been reported by a few studies. P2Y_12_ has been found in glioma and astrocytoma cells [61,62]. P2Y_12_ expression has been also described in breast cancer cell lines and was upregulated by cell treatment with cisplatin [63]. In those cells, co-administration of P2Y_12_ inhibitor with cisplatin resulted in significantly higher cytotoxic response [63]. 

## 3. P2Y_12_ Antagonists 

### 3.1. Thienopyridines

As previously described, P2Y_12_ activation by ADP has a fundamental role in thrombus formation and stabilization. Hence, antiplatelet agents inhibiting P2Y_12_ form a cornerstone of therapy for patients at risk of major adverse cardiovascular events (MACE), especially those with acute coronary syndrome undergoing percutaneous coronary intervention, as well as in the secondary prevention of cardiovascular events [33]. Besides their antithrombotic benefits, P2Y_12_ inhibitors also carry a bleeding risk, in both procedural and non-procedural settings [64,65,66].

Pharmacological description of P2Y_12_ inhibitors is essential to better understand their potential role in the modulation of inflammation, whether by a platelet dependent and/or an independent mechanism.

The P2Y_12_ inhibitors involve two classes of drugs: the thienotetrahydropyridines or thienopyridines (clopidogrel, prasugrel, and ticlopidine) and the nucleoside–nucleotide derivatives (cangrelor and ticagrelor) [33,67,68,69]. Pharmacological properties of P2Y_12_ inhibitors are summarized in Table 1.

Thienopyridines are orally administered inactive prodrugs that develop their antiplatelet effects only after in vivo metabolism to active metabolites [6]. Active metabolites of thienopyridines have a very short half-life and selectively interact with P2Y_12_ by forming a disulfide bond with cysteine-97 and then irreversibly inhibit ADP-induced activation [6]. Consequently, the inhibitory effect of thienopyridines on platelets lasts for the lifespan of a circulating platelet, which is 8 to 12 days on average [6].

The active metabolites of thienopyridines, and particularly for clopidogrel, are produced by a complex multistep hepatic cytochrome P450 (CYP450) dependent activation [6,77]. Although clopidogrel was shown to exhibit a beneficial effect in the prevention of MACE, it suffered from many drawbacks including slow onset of action, high variability of response, drug resistance due to CYP450 polymorphisms, moderate inhibition of platelet aggregation (33–64%) and drug interactions [6,67,68,77,78]. This led to the development of third generation thienopyridine prasugrel. Unlike ticlopidine and clopidogrel, prasugrel is first metabolized by an intestinal esterase and then undergoes a single step CYP450 dependent activation [6,79]. This allows a faster onset of action, more potent inhibition of platelet aggregation (50–80%), fewer resistance and drug interactions [68,77,80,81]. Consequently, prasugrel is associated with a higher bleeding risk [67].

### 3.2. Direct P2Y_12_ Inhibitors

By contrast, several reversible direct-acting P2Y_12_ receptor antagonists were developed using adenine-nucleotides (ADP or ATP) as the primary structure, that do not require hepatic metabolism [6,33,67,68,69]. Altogether, this pharmacological approach allows a faster, more potent and more predictable platelet inhibitory effect than thienopyridines [6,33].

Ticagrelor is an oral direct-acting P2Y_12_ inhibitor belonging to the cyclopentyl-triazolo-pyrimidines class. It has a rapid onset of action (2h) and produces a dose-dependent and important inhibition of platelet aggregation (60–90%) [68,71,73]. However, whether ticagrelor exerts a competitive or noncompetitive inhibition of ADP-induced P2Y_12_ activation is still debated [82,83]. Ticagrelor has one active metabolite (AR–C124910XX) produced by hepatic metabolism, which is at least equipotent and accounts for a third of its antiplatelet effects [73,84]. Even if the offset of action is significantly faster than prasugrel, the more potent platelet inhibitory effect induces a rate of offset still equivalent to clopidogrel, with platelet aggregation returning to pretreatment values after 5 days [68,71]. 

Ticagrelor exhibits PY12-independent pleiotropic effects. In addition to anti-P2Y_12_ activity, it has been shown to inhibit equilibrative nucleoside transporter 1 (ENT-1) which is a ubiquitous membrane transport protein, notably responsible for adenine uptake by erythrocytes and platelets [85,86,87]. Ticagrelor inhibits adenine uptake in vitro, inducing increased biological effect of exogenous adenosine and increased plasma levels and biological effects of endogenous adenosine [87]. High plasma levels of adenosine can induce platelet inhibition and coronary vasodilatation, reduce inflammatory response and improve ischemia/reperfusion injuries [87]. Thus, ENT-1 inhibition by ticagrelor may account for some of its biological effect, including inhibition of platelet activation. However, ticagrelor-induced increase in adenosine plasma levels in vivo remains very controversial [87,88,89].

It was also established that, together with an inhibition of platelet ENT1, ticagrelor can act as an inverse agonist at the P2Y_12_ receptor [86]. This inverse agonist effect was further investigated in the study of Garcia, supporting a new concept of P2Y_12_ receptor constitutive Gi/o-dependent signaling [90]. This effect was inhibited by ticagrelor but not by the thienopyridine inhibitors and led to an increase of cAMP-dependent signaling pathway compared to resting condition [90]. Moreover, Reiner et al. showed that ticagrelor exhibited endothelial-specific antithrombotic properties by reducing tissue factor expression and activity, independently of P2Y_12_ and ENT-1 [28]. Finally, a recent study by Lancellotti et al. demonstrated that ticagrelor had a bactericidal activity against antibiotic-resistant gram-positive bacteria in vitro and inhibited biofilm growth and dissemination of bacteria on a mouse model of implanted Staphylococcus aureus–preinfected disks [91]. 

Cangrelor is the only intravenous P2Y_12_ inhibitor. It is a direct-acting ATP analogue that selectively and reversibly binds to the P2Y_12_ receptor in a dose-dependent manner, without requiring metabolism. It achieves a high inhibition of platelet aggregation (>80%) with a rapid onset of action (2 min) and a very fast offset of action (30 to 60 min) due to a half-life of 3 to 6 minutes [76]. It is therefore a therapeutic option for patients with coronary artery disease (CAD) undergoing percutaneous coronary intervention (PCI), and to maintain P2Y_12_ inhibition when oral therapy is interrupted for any reason [92,93].

## 4. Effects of P2Y_12_ Inhibitors on Inflammation: Possible Molecular and Cellular Mechanisms

As described above, ADP-mediated activation of P2Y_12_ seems to be a common activating pathway in many inflammatory and immune cell types including platelets, leukocytes and dendritic cells. 

Hemostasis and inflammation are intimately linked, inducing and amplifying each other and this interconnection contributes to many pathological situations including sepsis, acute lung injury, autoimmune diseases, tumorigenesis and metastasis.

Hence, P2Y_12_ inhibitors might modulate inflammation through many different cellular and molecular mechanisms, summarized in Figure 2 and further detailed in this section. 

### 4.1. Thrombin Generation

Thrombin generation is a critical step during in vivo thrombogenesis that initiates the formation of fibrin clots and platelet activation. Besides, thrombin can mediate direct effects on inflammatory and immune cells through activation of protease-activated receptors (PARs), and notably PAR-1 [94,95]. Thrombin has been shown to induce the secretion of proinflammatory cytokines interleukin-1 (IL-1), interleukin-6 (IL-6), tumor necrosis factor α (TNFα), and monocyte chemotactic protein MCP-1 from different cell types including vascular endothelial cells and monocytes [96,97,98]. It also enhances angiogenesis by stimulating the expression of angiogenic growth factors including platelet derived growth factor (PDGF) and vascular endothelial growth factor (VEGF) [99,100]. Finally, thrombin has been shown to support T-cell proliferation and to induce leukocyte recruitment through PAR-1 activation [94,97]. 

Interplay between thrombin generation and inflammation involves positive feedback as leukocyte activation can also trigger thrombin generation, mainly through the release of neutrophil extracellular traps (NETs) and the expression of tissue factor by monocytes, macrophages and leukocyte-derived microparticles [95,101]. 

As described above, P2Y_12_ plays a role in thrombin generation and in thrombus formation in vitro. Accordingly, P2Y_12_ antagonism has been shown to inhibit thrombin generation through the decreased formation of coated-platelet in vitro and in vivo [50,51,102,103]. Clopidogrel can reduce ex vivo thrombin generation triggered by low concentrations of TF in rodent platelet rich plasma [104]. In a murine experimental endotoxemic model, clopidogrel was recently shown to reduce TF expression on leukocytes [105]. Finally, ticagrelor inhibited TF expression and activity in human aortic endothelial cells, independently of P2Y_12_ and ENT-1, and decreased thrombosis and endothelial TF expression in a photochemical mouse model of arterial thrombosis [28].

### 4.2. Release of Inflammatory Mediators

Inflammatory mediators, including cytokines, chemokines and growth factors are key modulators of acute and chronic inflammatory processes. Most of them, including IL-1, IL-6, IL-8 and TNFα are essentially produced by macrophages, monocytes and lymphocytes. They play a critical role in the positive feedback between hemostasis and inflammation [95,106]. Indeed, they induce proliferation, activation and recruitment of leukocytes, but also TF expression on cell surface of mononuclear and endothelial cells, as well as platelet activation [95,106,107,108]. 

Upon activation, platelets secrete a great amount of soluble mediators from their granules, with potent procoagulant and proinflammatory effects [109,110]. Cytokines and chemokines released by platelets include IL-1β, CD40L, platelet factor 4 (PF4, CXCL4), RANTES (CCL5), βTG (CXCL7), and IL-8 (CXCL8) [106,109,110]. These mediators play an important role in paracrine activation of platelets and in immune cell activation, proliferation and chemotaxis [106]. Notably, soluble CD40L activates platelets, induces IL-6 of and monocyte chemoattractant protein (MCP-1) secretion and enhances TF expression by monocytes, leading to increased thrombin generation [111,112].

P2Y_12_ inhibition by clopidogrel or by gene knockout in murine models of abdominal sepsis or lipopolysaccharide (LPS)-induced inflammation has been shown to decrease proinflammatory mediator levels in plasma, notably IL-6, TNFα, CCL4 (MIP-1β) and IL-1β [48,105,113]. This is consistent with the reduction in IL-6 and TNFα levels, upon clopidogrel treatment, in a rat model of LPS-induced inflammation [114]. In a human experimental model of intravenous LPS-induced inflammation, ticagrelor and clopidogrel were associated with a significant reduction of IL-6, TNFα and CCL2, and an additional reduction of IL-8 was obtained with ticagrelor [115]. However, anti-inflammatory effect of clopidogrel is inconsistently reported in clinical trials enrolling patients with acute coronary syndromes or stable coronary artery disease, with a reduction of CRP and TNFα [116]. Still, the recently published double-blinded randomized XANTHIPPE study comparing ticagrelor to placebo in pneumonia, demonstrated a significant reduction in plasma IL-6 levels in the ticagrelor group [113].

### 4.3. Platelet-Leukocyte Interactions and Formation of Neutrophil Extracellular Traps (NETs)

The pathophysiology of platelet-leukocyte interaction has recently been extensively reviewed by Rossaint et al. [117]. Resting circulating platelets act as sentinels in the blood. Upon activation, degranulation and membrane expression of platelet receptors enable physical interactions between platelets and leukocytes, especially neutrophils, forming so-called platelet-leukocyte aggregates (PLAs) [117,118,119]. This interaction is critical in both thrombosis and inflammatory situations. It allows leukocyte activation and recruitment to sites of inflammation and induces the release of proinflammatory mediators, reactive oxygen species (ROS) and neutrophil extracellular traps (NETs) by neutrophils [117,118,119].

The first interaction between platelets and leukocytes is established between P-selectin (CD62P) on activated platelets and PSGL-1, constitutively expressed on the surface of neutrophils, monocytes, dendritic cells, and subclasses of lymphocytes. CD62P/PSGL-1 interaction induces the activation of αMβ2 (Mac-1) integrin on leukocyte surface. Platelet leukocyte interaction is then stabilized by direct-binding of Mac-1 to platelet GPIbα and indirect binding to activated platelet αIIbβ3 (GPIIbIIIa) via fibrinogen [117,118,119].

NETs are large, web-like extracellular chromatin structures that are released by neutrophils upon various activating stimuli, including exogenous microorganisms (bacteria, fungi, virus, and parasites), immune complexes and activated platelets [120]. NETs can trap, neutralize and kill microorganisms, but also activate platelets and promote coagulation [121]. They are considered as critical players in many inflammatory conditions, especially sepsis-induced organ injuries, autoimmunity, tumorigenesis and metastasis [120]. NETs also contribute to thrombogenesis by forming a mesh with platelets and fibrin and accumulating coagulation factors such as TF [118,120,122].

The crosstalk between activated platelets and neutrophils is a critical regulator of NETosis [123,124]. Platelet-neutrophil direct binding is an essential step in platelet-driven NETosis and is mainly mediated by P-selectin/PSGL-1 and GPIbα/Mac-1 interactions [120,123,124,125]. NET formation is further stimulated by platelet-released soluble mediators including high mobility group box 1 (HMGB1), PF4 and RANTES [120,123,124,125].

P2Y_12_ inhibition by clopidogrel, prasugrel, ticagrelor and cangrelor has been consistently associated with inhibition of platelet-leukocyte interaction and platelet P-selectin expression in animal models of inflammation and in vitro studies [48,105,126,127,128,129]. In a human experimental model of intravenous LPS-induced inflammation, both ticagrelor and clopidogrel were associated with a significant reduction of platelet-monocyte aggregates [115]. Moreover, reduction of PLAs and P-selectin expression by clopidogrel was reported in clinical studies of patients with acute coronary syndromes and atherosclerotic vascular disease [130,131]. Finally, the recent XANTHIPPE study demonstrated a reduction in PLAs in the ticagrelor group compared to placebo [113]. To date, no study reported a significant modulation of NETosis by P2Y_12_ inhibitors, although ticagrelor was recently shown to reduce in vitro NET-induced platelet aggregation, secretion and expression of P-selectin [113,121].

### 4.4. Adenosine Mediated Effects

The mechanisms of adenosine signaling on immune cells has been extensively reviewed in a recent work by Vigano et al. [132]. Adenosine exerts a powerful modulatory effect on inflammation and innate immune responses through the activation of four different receptors (A1, A2A, A2B and A3) which are expressed in the majority of immune cells, including neutrophils, lymphocytes, macrophages, mast cells and dendritic cells. Adenosine principally induces an anti-inflammatory phenotype characterized by a decreased ability to release inflammatory mediators, an inhibition of immune cell activation, proliferation and chemotaxis. Therefore, adenosine acts as a regulator of immune cells that aims at preserving host integrity and promotes the resolution of inflammation. Moreover, adenosine is a potent inhibitor of platelet aggregation and adenosine receptor agonists have been shown to potentiate antiplatelet effects of P2Y_12_ antagonists [87,133,134]. Thus, adenosine might have a beneficial effect in the management of severe inflammatory disorders including sepsis.

As described above, ticagrelor has been shown to increase plasma levels of adenosine by inhibiting ENT-1. To date, no study demonstrated an adenosine-mediated anti-inflammatory effect of ticagrelor, although ticagrelor was shown to increase the in vitro inhibitory effect of exogenous adenosine on platelet aggregation [87].

## 5. Current Evidence for P2Y_12_ Inhibition in Clinical Inflammatory Diseases and Syndromes

P2Y_12_ is now considered as a potential target in several inflammatory diseases, including sepsis, asthma, atherosclerosis and cancer. Main clinical studies evaluating the effects of P2Y_12_ inhibition in those inflammatory diseases are reported in Table 2 and will be further discussed in this section.

### 5.1. Sepsis and Sepsis-Induced Acute Lung Injury (ALI)

Sepsis is defined as a life-threatening organ dysfunction that is caused by a dysregulated systemic inflammatory and immune host response to infection [145]. The mechanisms underlying the excessive inflammation in sepsis have been extensively reviewed by van der Poll et al. [146]. To summarize, sepsis-induced organ dysfunctions result from the interplay between uncontrolled activation of the complement, coagulation, and inflammatory systems. Platelet-mediated inflammatory response is now recognized as a critical player in the pathophysiology of sepsis, and is especially involved in sepsis-induced experimental acute lung injury (ALI) and clinical life-threatening acute respiratory distress syndrome (ARDS) through several mechanisms, including pathogen sensing, release of inflammatory mediators, recruitment and activation of immune cells and thrombosis [109,147,148,149,150,151]. 

First report of the potential benefits of P2Y_12_ inhibition in sepsis came from observational studies of patients receiving antiplatelet agents. A study of 224 consecutive patients admitted for community acquired pneumonia showed lower use of intensive care unit and shorter stay in hospital in patients receiving antiplatelet agents (aspirin and/or thienopyridines) for at least 6 months compared with age-matched controls [135]. More evidence came from a post hoc analysis of the PLATO trial, in which ticagrelor was shown to significantly reduce death from vascular causes, myocardial infarction, or stroke compared to clopidogrel in 18,624 patients with ACS [152]. This analysis on 18,421 patients revealed that, compared to clopidogrel, ticagrelor was associated with a lower mortality risk following pulmonary events and sepsis in acute coronary syndrome [136]. Finally, a recent observational study including 683,421 patients hospitalized for sepsis, showed a lower risk of mortality in patients receiving antiplatelet agents before admission [137]. Using a nested-control study design in 372,748 patients, they showed that both aspirin and P2Y_12_ inhibitors were associated with a lower adjusted risk of mortality [137].

Those findings were further confirmed in experimental animal models of endotoxemia and abdominal sepsis. In a rat model of LPS-induced endotoxemia, clopidogrel inhibited IL-6 and TNFα secretion and reduced lung and liver histological injuries [114]. Liverani et al. demonstrated in a mouse model of intra-abdominal sepsis that clopidogrel-treated and P2Y_12_ null mice were refractory to sepsis-induced lung injury and exhibited a decrease in platelet activation, platelet-leukocyte aggregation and release of inflammatory cytokines [48]. Those results were confirmed in another mouse model of intra-abdominal sepsis in which ticagrelor reduced histological findings of sepsis-induced lung injury, pulmonary infiltration of neutrophils, formation of PLAs and platelet activation [126]. To our knowledge, only one experimental study on LPS-treated mice demonstrated a benefit of ticagrelor on mortality [113].

The XANTHIPPE trial (Examining the Effect of Ticagrelor on Platelet Activation, Platelet-Leukocyte Aggregates, and Acute Lung Injury in Pneumonia), was the first double-blinded placebo-controlled randomized study to evaluate the effect of ticagrelor on inflammation, platelet activation and lung function in 60 patients with community or hospital-acquired pneumonia [113]. Ticagrelor administration to patients within 48h of pneumonia diagnosis demonstrated anti-inflammatory effect with reduced PLAs in circulation, lowered IL-6 levels and improved lung function with a decrease in supplemental oxygen requirements [113].

### 5.2. Asthma

Platelet activation is strongly involved in the pathogenesis of allergic asthma, including bronchial hyperresponsiveness and airway wall inflammation and remodeling [153,154,155]. Among proposed mechanisms, platelet P2Y_12_ pathway seems to play an important role. In a study of murine leukotriene E4-induced asthma, clopidogrel treatment and P2Y_12_ gene knock-out reduced IL-13 levels and airway eosinophilia and inflammation [20]. The underlying mechanism remains unknown. Two randomized trials evaluated the effect of P2Y_12_ inhibition in patients with asthma. In a double-blind placebo-controlled crossover trial of prasugrel in 40 patients with aspirin-exacerbated respiratory disease, prasugrel did not attenuate aspirin-induced symptoms and failed to decrease PLA levels and mast cell activation [138]. Finally, the PRINA (effect of prasugrel on bronchial hyperreactivity and on markers of inflammation in patients with chronic asthma) trial, in which 26 asthmatic patients were randomly and blindly allocated to prasugrel or placebo for 15 days followed by a 15-day wash-out and a cross-over, showed that P2Y_12_ inhibition by prasugrel significantly decreased airway hyperresponsiveness with improvement of forced expiratory volume [139]. 

### 5.3. Atherosclerosis

Atherosclerosis is a chronic inflammatory vascular disease involving cellular and molecular interactions between platelets, endothelial cells, VSMCs and monocytes, interactions that deeply involve the P2Y_12_ receptor [156,157,158].

The role of P2Y_12_ in atherosclerosis has been well documented in experimental models. In a mouse model of atherosclerosis by Apolipoprotein E (ApoE) gene knock-out, genetic P2Y_12_ deletion was associated with reduced lesion area, increased fibrous content at the plaque site and decreased inflammatory cell infiltration [159]. Likewise, clopidogrel and ticagrelor limited the progression of late atherosclerotic lesion and promoted plaque stability in ApoE knockout mice [160,161,162,163]. Unfortunately, differential analysis of the role of platelet and non-platelet P2Y_12_ receptors was not possible with these studies. Therefore, a study using P2Y_12_ gene knockout and bone marrow transplantation in ApoE null mice demonstrated that, compared to vessel wall P2Y_12_, platelet P2Y_12_ had no effect on early atheroma formation, implying additional role of VSMC P2Y_12_ receptors [164]. Further studies are necessary to clarify the extent to which vessel wall and platelet P2Y_12_ influence early and late atherogenesis.

### 5.4. Cancer—Tumor Growth and Metastasis

The recent advances in the knowledge of cancer have shown that the biology of cancer has evolved from a tumor-centered view to a concept that places cancer cells within a cell network, including fibroblasts, vascular cells and inflammatory immune cells, that form the tumor microenvironment (TME) [165]. Modulation of inflammation within the TME is a critical player during all stages of tumorigenesis and involves reciprocal interactions between cancer cells and surrounding inflammatory cells. These interactions can promote tumor growth by direct proliferative effect on tumor cells and by inducing immunosuppression [165]. 

The role of platelets in all steps of tumorigenesis including tumor growth, tumor cell extravasation and metastasis has been extensively studied and is now well-recognized [9,166]. Activated platelets not only secrete PF4 but also lipids, microRNAs and numerous growth factors, including TGFβ (transforming growth factor β) and VEGF, which favor tumor cell proliferation, metastasis and angiogenesis. Moreover, the interaction between platelets and cancer cells is largely described in the literature as a mechanism favoring tumor- cell induced platelet activation (TCIPA), and inducing a “shield” of platelets coating the tumor cells and protecting them from the immune response [9].

#### 5.4.1. In Vitro and Preclinical Studies on P2Y_12_ Inhibition in Cancer 

Consequently, most of published data have evidenced a net anti tumoral effect of antiplatelet agents, including aspirin, cilostazole and P2Y_12_ inhibitors. The rationale to focus on anti-P2Y_12_ agents came recently first from the discovery of ADP secretion by tumor cells. For instance, Cho et al. nicely showed the secretion of ADP by ovarian cancer cells [167]. Therefore, platelets can be activated directly by tumor cells via cell-to-cell interactions, and indirectly via ADP secreted by tumor cell. Second is the expression of P2Y_12_ receptors in cells other than platelets, such as VSMCs and tumor cells such as breast cancer cell lines [27]. In Cho’s study, the use of ticagrelor reduced tumor growth by 60% compared to aspirin and 75% versus placebo, inhibited proliferation as assessed by Ki67 positivity, and increased apoptosis of ovarian cancer cells. In P2Y_12_−/− mice, the growth of ovarian tumors was reduced by over 85% compared to wild-type animals, but not in P2Y_1_−/− mice, pointing to the essential role of P2Y_12_ between the two ADP receptors. Finally, the invalidation of ectopyrase gene (CD39, which catabolizes ADP) in cancer cells increased platelet-related cancer cell proliferation.

This finding is not restricted to ovarian tumor cell lines. Gareau et al. showed the beneficial effect of ticagrelor in limiting the interaction between platelets and human mammary carcinoma cells [168]. 

In other preclinical studies, ticagrelor used at clinical dose (10 mg/kg) inhibited metastasis and improved survival in a mouse model of melanoma metastasis [169]. 

In a Lewis lung carcinoma spontaneous metastatic mouse model, P2Y_12_ deficiency reduced pulmonary metastasis and reduced the ability of platelets to secrete active TGFβ1, thus limiting epithelial to mesenchymal transition and invasiveness of tumor cells [170]. Same findings were observed with B16 melanoma metastasis model.

Consequently, targeted therapies start to emerge in the aim to prevent platelet-tumor cell interactions. The tumor-homing peptide Cys-Arg-Glu-Lys-Ala (CREKA) that targets fibrin-fibronectin complexes found on the tumor stroma and vessel wall was linked to ticagrelor. CREKA-ticagrelor inhibited platelet-induced migration of 4T1 tumor cells and prevented tumor-platelet interaction. In vivo, it suppressed lung metastasis in a mouse model injected with breast cancer cells [171]. This finding opens the way to innovative antimetastatic agents.

#### 5.4.2. Are anti P2Y_12_ Agents Protective or at Risk for Increased Cancer-Related Events?

Numerous studies analyzing the potential effect of antiplatelet agents on cancer converge to a protective effect of low dose aspirin on cancer-related mortality. For instance, in the meta-analysis of Algra and Rothwell including 17 studies, regular use of aspirin was protective from colorectal cancer ([OR] 0.62, 95% CI 0.58–0.67, *p* < 0.0001) [172]. Given that COX1 inhibition is predominant over COX2 inhibition at low dose aspirin, this effect is likely to result primarily from the antiplatelet effect of aspirin. 

By contrast, results from randomized clinical trials on anti P2Y_12_ agents are conflicting. Whereas CAPRIE and CHARISMA studies of clopidogrel versus aspirin did not report increased cancer development, some data from trials using prolonged anti P2Y_12_ treatment showed increased rates of cancer-related mortality [173,174]. TRITON-TIMI 38 trial of prasugrel compared to clopidogrel on top of aspirin for 6 to 15 months showed a significantly accelerated cancer progression and increased risk of cancer death in the prasugrel group, particularly with breast, colorectal and prostate cancers [175]. One explanation for this apparent paradoxical effect was that the more potent antiplatelet effect of prasugrel brought more events to medical attention and to an increased number of diagnosed cancers. However, results were different in the TRILOGY trial with no difference in cancer frequency between clopidogrel and prasugrel groups after a median follow-up of 17 months [176]. Clopidogrel and ticagrelor given more than 12 months after drug-eluting stenting in the DAPT trial showed a significant increase in cancer-related deaths [177]. However, deaths related to cancer in this study were relatively low in number. Also concerning ticagrelor, PEGASUS-TIMI 54 trial showed an enhanced cancer risk of ticagrelor administered beyond 1 year, whereas PLATO was negative [152,178,179]. Interestingly Raposeiras-Roubin et al. performed a retrospective study on 4229 consecutive acute coronary syndrome patients with a median follow up of 46 months [140]. They found that ticagrelor resulted in a lower cancer risk than clopidogrel without difference between clopidogrel and prasugrel. Noteworthy, only 311 patients were diagnosed with cancer during the follow up (incidence 2.1 per 100 people per year) and ticagrelor-receiving population was 459 versus 3530 with clopidogrel.

Overall, these clinical randomized trials do not include an untreated comparator arm, and are not powered to detect differences in cancer-related events or mortality. Consequently, the Food and Drug Administration (FDA) reported a two trial-level that “rejected the hypothesis of cancer association in patients on dual anti platelet therapy with clopidogrel, that is, the adverse mortality findings in the DAPT trial were not confirmed” [180]. Moreover, the FDA Adverse Event Reporting system is probably unreliable for adequate assessment of cancer risk during antiplatelet treatment as associated cancers might be unreported and/or missed [181]. 

The evidence for no cancer risk with P2Y_12_ inhibitors mostly stems from meta-analysis and cohort studies. The meta-analysis of Kotronias et al included nine studies with more than 282,000 participants [182]. When compared with standard aspirin or placebo, the thienopyridines clopidogrel and prasugrel were not associated with cancer mortality and event rate. The study concluded that there was insufficient evidence to suggest an association between thienopyridine exposure and increased risk of cancer event rate or mortality.

The question of the duration of treatment was also addressed in cohort studies. Leader et al showed a lower risk of cancer in subjects exposed to aspirin compared to non-users, with or without clopidogrel, on long-term follow-up [141]. In a large cohort of 10,359 colorectal cancer, 17,889 breast cancer, and 13,155 prostate cancer patients, Hicks et al evaluated the post-diagnostic use of clopidogrel and cancer-specific mortality during an average follow-up of 5 years [142]. Overall, there was no increase in the rate of cancers in patients receiving clopidogrel, after adjustment for potential confounders. Finally, the meta-analysis of Elmariah et al including more than 48,000 patients from six randomized trials confirmed the absence of impact of prolonged clopidogrel on top of aspirin on mortality or cancer [143]. More recently, Rodriguez-Miguel et al showed in 15,491 cases of colorectal cancer versus 60,000 controls, that low-dose aspirin was associated with a reduced risk of colorectal cancer incidence in patients receiving treatment for more than one year [144]. Same reduction of 20 to 30% was found for clopidogrel alone or in combination with aspirin. In short-term users, there was on the contrary an increased risk for patients on clopidogrel and aspirin. Again, the hypothesis raised was an increased incidence of gastro-intestinal bleedings that led to a greater number of colonoscopies and early diagnosis.

Altogether, if it is challenging to compare the effects of antiplatelet agents on cancer-related death in studies designed to analyze adverse cardiovascular-related events, a head-to-head comparison between molecules is also questionable because their pharmacology differs. The thienopyridine clopidogrel has a less predictable effect than prasugrel or ticagrelor. Clopidogrel is indeed less effective in a subset of around 30% of patients, especially those carrying loss-of-function 2C19*2 variant [78]. Besides, ticagrelor not only has more predictable pharmacokinetics than clopidogrel but has been shown to exert an anti-inflammatory effect due to an increased adenosine extracellular concentration. Finally, most studies have investigated the effect of anti P2Y12 agents on top of aspirin and it is difficult to conclude for one drug over the others. 

## 6. Conclusions

P2Y_12_-mediated nucleotide signaling is now considered to be a critical player in inflammatory response. Besides platelets, P2Y_12_ is expressed in many immune and vascular cells and has been shown to modulate inflammatory processes, including the release of inflammatory mediators, platelet–leukocyte interactions, and thrombosis. In addition to their antithrombotic properties, P2Y_12_ inhibitors can therefore be considered to have valuable pharmacological targets for inflammation, and the beneficial effects of anti-P2Y_12_ drugs have been reported for several experimental and clinical inflammatory diseases, including sepsis, acute lung injury, asthma, atherosclerosis, and cancer. Still, evidence is missing on the exact role of nonplatelet P2Y_12_ receptors and on how to distinguish between the P2Y_12_-mediated and adenosine-mediated effects of ticagrelor. More prospective controlled clinical studies are warranted to determine the potential benefits of P2Y_12_ inhibition therapy for inflammatory diseases, and pharmacological strategies should be further developed to overcome potential P2Y_12_-associated bleeding risks.

## Figures and Tables

**Figure 1 ijms-21-01391-f001:**
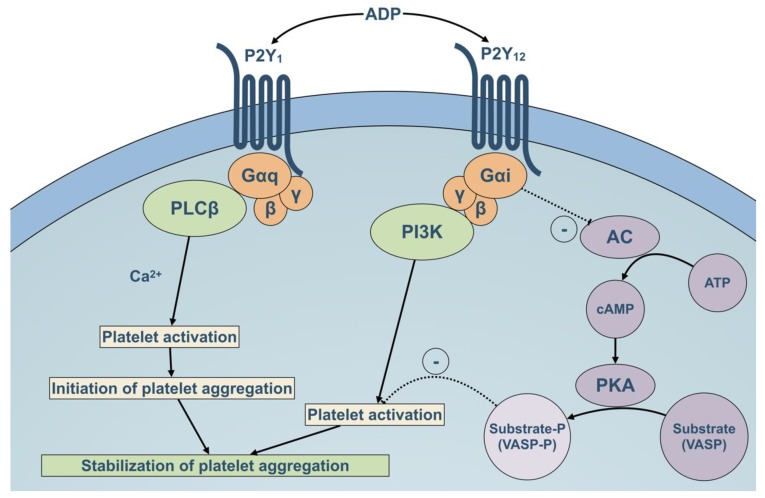
Schematic illustration of the platelet P2-mediated ADP signaling pathway and its role in platelet activation. ADP can induce platelet activation by interacting with two G-coupled platelet receptors, P2Y_1_ and P2Y_12_, and contributes to the stabilization of platelet aggregation induced by other strong agonists such as thrombin and TXA2. ADP binding to P2Y_1_ induces a mobilization of calcium ions through the stimulation of phospholipase C and therefore initiates early reversible aggregation. ADP binding to P2Y_12_ induces PI3K activation through the recruitment of βγ subunits of Gi and inhibition of adenylate cyclase through the recruitment of αi subunit, therefore decreasing cAMP levels and leading to impaired PKA activation. These pathways ultimately lead to platelet activation, αIIbβ3 activation and stabilization of platelet aggregates. AC: adenylate cyclase; ADP: adenosine diphosphate; ATP: adenosine triphosphate; PI3K: phosphoinositide-3-kinase; PKA: protein kinase A; PLC: phospholipase C; VASP: vasodilator-stimulated phosphoprotein.

**Figure 2 ijms-21-01391-f002:**
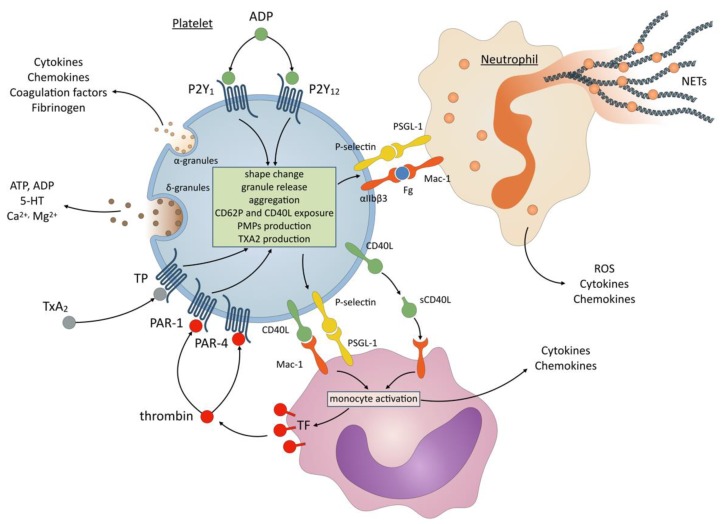
Schematic overview of the role of platelet P2Y_12_ activation in inflammation. ADP binding to P2Y_12_ initiates platelet activation and amplifies granule secretion and aggregation induced by ADP-mediated P2Y_1_ stimulation and by strong agonists such as thrombin (PAR-1 and PAR-4 receptors) and TXA_2_ (TP receptor). Activated platelets release alpha and dense granules contents, including cytokines, chemokines, coagulation factors and platelet agonists. Cytokines, chemokines and soluble CD40 ligand (sCD40L) can recruit and activate leukocytes, especially neutrophils and monocytes. Expression of P-selectin and CD40L at the surface of activated platelets allow interaction between platelets and leukocytes which is a critical step in platelet-mediated inflammation. P-selectin cross-links platelets and leukocytes through its corresponding ligand P-selectin glycoprotein ligand-1 (PSGL-1) present on monocytes and neutrophils. Platelet-leukocyte aggregates are further stabilized by numerous additional receptor/ligand pairs, especially leukocyte Mac-1 that recognizes platelet CD40L and fibrinogen (Fg) bound to platelet αIIbβ3. Activation by platelet interaction or soluble mediators stimulates monocyte cytokine and chemokine production, expression of tissue factor and thrombin generation. Platelet-bound neutrophils produce inflammatory mediators including reactive oxygen species (ROS), and release neutrophil extracellular traps (NETs) that play a critical role in many inflammatory conditions, especially sepsis-induced organ injuries, autoimmunity, tumorigenesis and metastasis.

**Table 1 ijms-21-01391-t001:** Pharmacological properties of P2Y_12_ receptor inhibitors.

Drug	Ticlopidine	Clopidogrel	Prasugrel	Ticagrelor	Cangrelor
Target	P2Y_12_	P2Y_12_	P2Y_12_	P2Y_12_ ENT-1	P2Y_12_
P2Y_12_ receptor binding	Irreversible	Irreversible	Irreversible	Reversible	Reversible
Route of administration	Oral	Oral	Oral	Oral	Intravenous
Metabolism	Prodrug CYP450	Prodrug Esterase CYP450	Prodrug Intestinal Esterase CYP450	Direct-Acting and CYP450	Direct-Acting Dephosphorylation
Time to maximum IPA ^1^	3–4 days	4–5 h	2–4h	2–4 h	2 min
Steady-state IPA ^2^	20–30%	33–64%	43–73%	82–95%	>80%
Offset of action ^3^	11–13 days	5–7 days	7–10 days	3–5 days	30–60 min

CYP450: cytochrome P450; IPA: inhibition of platelet aggregation; ^1^ after a loading dose; ^2^ percentage inhibition of platelet aggregation measured by light transmission aggregometry; ^3^ based on return of platelet aggregation and/or bleeding time to baseline values; [67,68,69,70,71,72,73,74,75,76].

**Table 2 ijms-21-01391-t002:** Clinical studies investigating anti-inflammatory effects of P2Y_12_ antagonists.

Study	Type of Study	Condition	Antiplatelet Drugs Evaluated	Effects of P2Y_12_ Antagonists
Winning et al. [135]	Observational 224 patients	CAP	ASA and thienopyridines	Lower use of ICU and shorter stay in hospital (thienopyridines plus ASA or thienopyridines alone
Storey et al. [136]	Observational post hoc analysis18,421 patients	ACS	Ticagrelor vs clopidogrel	Lower mortality risk following pulmonary events and sepsis
Tsai et al. [137]	Observational 683,421 patients	Sepsis	ASA and thienopyridines	Lower risk of mortality
XANTHIPPE Sexton et al. [113]	RCT 60 patients	Pneumonia (CAP, HAP)	Ticagrelor vs placebo	Reduced PLAs and IL-6 levels improved. Decreased supplemental oxygen requirements
Laidlaw et al. [138]	Crossover RCT 40 patients	AERD	Prasugrel vs placebo	No effect on clinical or inflammatory parameters
PRINA Lussana et al. [139]	Crossover RCT 26 patients	Chronic asthma	Prasugrel vs placebo	Decreased airway hyperresponsiveness
Raposeiras-Roubin et al. [140]	Observational 4229 patients	ACS	Ticagrelor, Clopidogrel and Prasugrel	Lower cancer risk ticagrelor vs clopidogrelNS clopidogrel vs prasugrel
Leader et al. [141]	Observational 3479 patients	ACS	ASA and clopidogrel	Lower cancer risk (clopidogrel plus ASA or clopidogrel alone)
Hicks et al. [142]	Observational 41,403 patients	Cancer (breast, colorectal, prostate)	Clopidogrel	No increase in cancer risk
Elmariah et al. [143]	Meta-analysis 48,000 patients	Cardiovascular and cerebrovascular disease	ASA and clopidogrel	No increase in cancer risk
Rodriguez-Miguel et al. [144]	Observational 75,491 patients	CRC	ASA and clopidogrel	Lower cancer risk (clopidogrel plus ASA or clopidogrel alone)

CAP: community acquired pneumonia, HAP: hospital acquired pneumonia, ASA: acetylsalicylic acid=aspirin, ICU: intensive care unit, ACS: acute coronary syndrome, RCT: randomized control trial, AERD: aspirin-exacerbated respiratory disease, NS: non-significant difference, CRC: colorectal cancer.

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
