# Peer review of "P2Y12 Inhibition beyond Thrombosis: Effects on Inflammation"

_ijms, 2020, doi:10.3390/ijms21041391_

Round 1

Reviewer 1 Report

            The manuscript by Mansour et al. is an excellent exposition of the involvement of P2Y12 receptors in inflammatory signaling and the effect of P2Y12 antagonists.  The latter are employed extensively for their anti-platelet and anti-thrombotic effects, and only more recently has it become apparent that they may also have a role in inflammation, sepsis, and even cancer.  I would suspect that most of the potential audience will be unaware of these latter observations.

            Although the manuscript is well written, there are multiple housekeeping issues including typographical errors, syntax errors, and poor word choice.  These issues are described below, and corrections will require little effort.

60—“antagonism” is better than “antagonization”. 147—Do the authors mean “adaptive immunity”? 160—Insert “of” before P2Y12. 191, 208, 214, 241—“metabolism” is better than “metabolization”. 194—“lasts” 201—“and then undergoes” is better. 215—“accounts” 232—“inhibited by ticagrelor” is better. 307—Delete “of”. 351—Change “recently” to “recent”. 472—“In Cho’s study” is better. 492—“opens” 517—Change “of” to “with”. 540—“confirmed” is better than “concluded to”. 553—Delete “a”. 566—“ticagrelor”

   Abbreviations: MACE—add “event”

                         VASP—add “phosphoprotein”

                         Consider adding CREKA, EMT, FDA, GPIIb/IIIa, PDGF, TCIPA

   In reference 37 first author’s name is not complete.

   In references 37, 50, 78, and 80 authors’ names are all capitalized.                  Change so all reference have same style.

   References 83 and 127 are identical.  Please correct and check for other             duplicates.

   In reference 166 who are the authors?

   In reference 173 it is uncertain who the authors are.

Author Response

We would like to thank the reviewer for careful and thorough reading of this manuscript and for the positive feedback.

We sincerely apologize for the multiple spelling or typographical errors (especially in the references). All the minor issues listed by the reviewer are now corrected.

Reviewer 2 Report

In the review article „P2Y12 inhibition beyond thrombosis: effects on inflammation” by Mansour et al., the Authors review the literature on P2Y12 receptor as therapeutic target, focusing on the importance of the receptor inhibition in the inflammation. The paper is very interesting and well-written and fills a gap in the current state of the knowledge since the majority of reviews obviously deal with a pharmacology of P2Y12 receptor in thrombosis and haemostasis.

Major comment:

In the section 4.4 Adenosine mediated effects, the Authors could also mention a role of adenosine in modulating the activation of blood platelets and a therapeutic potential of adenosine receptors agonists, especially in a dual strategy including the simultaneous use of P2Y12 antagonists (Boncler et al. Vascul Pharmacol. 2019:113:47-56; Wolska, Int J Mol Sci. 2019:20(21); Wolska et. al Molecules 2020, 25, 130)

Minor comments:

The reference list should be carefully checked and proofread. A couple of examples are listed below:

Ref 37. Names of the authors are superscripted

Ref. 50. Names of the authors are superscripted

Ref. 78. Names of the authors are superscripted

Ref. 80. Names of the authors are superscripted

Ref. 50. Names of the authors are superscripted

Ref. 173. „Food and drug” is missing in the beginning of the reference

Author Response

We would like to thank the reviewer for careful and thorough reading of this manuscript, for his positive feedback and for valuable comments, which help to improve the quality of this manuscript.

Reply to major comment:

As suggested by the reviewer, we included a mention of the modulatory effect of adenosine on antiplatelet effect of P2Y12 inhibitors in section 4.4 (371).

Reply to minor comments:

We sincerely apologize for the multiple spelling or typographical errors (especially in the references). Reference list was checked and corrected.

Reviewer 3 Report

This review by Mansour at al provides a very informative discussion on how P2Y12 can be considered as a targeted to control inflammation. The description of expression of P2Y receptors on various cell types and how its activation can facilitate interaction with other cell types are discussed in detail. A nice integration of all inhibitors of p2Y12 receptor in presented in Table 1.

A strength of the review is the inclusion of discussion on interplay of ADP receptor under different disease conditions, particularly in cancer. There are a few minor concerns that should be addressed to make it complete.

A description of how platelet-neutrophil interaction primes for NETosis should be included with references. Authors should discuss that use of inhibitors of ADP receptor can pose bleeding issues. This should be acknowledged as a limitation and can be included in conclusion.

Author Response

We would like to thank the reviewer for careful and thorough reading of this manuscript, for his positive feedback and for valuable comments, which help to improve the quality of this manuscript.

As suggested by the reviewer, we included a detailed description of platelet-driven NETosis in section 4.3 (345-349). P2Y12 inhibitors associated bleeding risk was added as a brief discussion in section 3 (177-178) and as a perspective in the conclusion (574-575).